# A Smart Cane Based on 2D LiDAR and RGB-D Camera Sensor-Realizing Navigation and Obstacle Recognition

**DOI:** 10.3390/s24030870

**Published:** 2024-01-29

**Authors:** Chunming Mai, Huaze Chen, Lina Zeng, Zaijin Li, Guojun Liu, Zhongliang Qiao, Yi Qu, Lianhe Li, Lin Li

**Affiliations:** 1College of Physics and Eletronic Engineering, Hainan Normal University, Haikou 571158, China; 20213085400008@hainnu.edu.cn (C.M.); zenglina@hainnu.edu.cn (L.Z.); lizaijin@hainnu.edu.cn (Z.L.); 068006@hainnu.edu.cn (G.L.); qzhl060910@hainnu.edu.cn (Z.Q.); quyi@hainnu.edu.cn (Y.Q.); 2College of Information Science and Technology, Hainan Normal University, Haikou 571158, China; 202213085400019@hainnu.edu.cn; 3Key Laboratory of Laser Technology and Optoelectronic Functional Materials of Hainan Province, Hainan Normal University, Haikou 571158, China; 4Hainan International Joint Research Center for Semiconductor Lasers, Hainan Normal University, Haikou 571158, China; lilianhehnnu@126.com

**Keywords:** smart cane, Jetson nano (B01), 2D LiDAR, RGB-D camera, laser SLAM, target recognition, cartographer, improved YOLOv5

## Abstract

In this paper, an intelligent blind guide system based on 2D LiDAR and RGB-D camera sensing is proposed, and the system is mounted on a smart cane. The intelligent guide system relies on 2D LiDAR, an RGB-D camera, IMU, GPS, Jetson nano B01, STM32, and other hardware. The main advantage of the intelligent guide system proposed by us is that the distance between the smart cane and obstacles can be measured by 2D LiDAR based on the cartographer algorithm, thus achieving simultaneous localization and mapping (SLAM). At the same time, through the improved YOLOv5 algorithm, pedestrians, vehicles, pedestrian crosswalks, traffic lights, warning posts, stone piers, tactile paving, and other objects in front of the visually impaired can be quickly and effectively identified. Laser SLAM and improved YOLOv5 obstacle identification tests were carried out inside a teaching building on the campus of Hainan Normal University and on a pedestrian crossing on Longkun South Road in Haikou City, Hainan Province. The results show that the intelligent guide system developed by us can drive the omnidirectional wheels at the bottom of the smart cane and provide the smart cane with a self-leading blind guide function, like a “guide dog”, which can effectively guide the visually impaired to avoid obstacles and reach their predetermined destination, and can quickly and effectively identify the obstacles on the way out. The mapping and positioning accuracy of the system’s laser SLAM is 1 m ± 7 cm, and the laser SLAM speed of this system is 25~31 FPS, which can realize the short-distance obstacle avoidance and navigation function both in indoor and outdoor environments. The improved YOLOv5 helps to identify 86 types of objects. The recognition rates for pedestrian crosswalks and for vehicles are 84.6% and 71.8%, respectively; the overall recognition rate for 86 types of objects is 61.2%, and the obstacle recognition rate of the intelligent guide system is 25–26 FPS.

## 1. Introduction

The problem of visual impairment is very common, and there is a growing trend. According to a 2015 survey of visual impairment, an estimated 253 million people worldwide suffer from it. Of these, 36 million are blind and 217 million suffer from moderate to severe visual impairments [1]. According to the World Outlook Report released by the World Health Organization (WTO) in 2019, at least 2.2 billion people in the world are visually impaired or blind [2]. By 2020, the number of blind people worldwide was estimated to have grown to 43.3 million [3]. It is estimated that 61 million people worldwide will be blind by 2050 [3]. According to the data from the official website of the Blind Association, China is the country with the largest number of blind people in the world, and the number of visually impaired people exceeds 17.3 million, of whom, 8 million are completely blind. About 1 in every 120 Chinese people suffers from visual impairment diseases, and about 450,000 additional visually impaired people exist in China every year. At present, due to the small number of guide dogs in China, most visually impaired people choose to move around with the help of a white cane. However, the traditional white cane is composed of a simple cane body and handle, which performs a singular function and has a limited range of detection. In recent years, with the continuous maturity of autonomous driving technology, robot autonomous navigation technology, object detection and recognition technology, and the increase in intelligent hardware, a new solution to the travel problem of the visually impaired has been provided.

Researchers around the world have made great contributions to laser and vision-sensing smart guide systems [4,5,6,7,8,9,10,11,12]. Laser and vision-sensing smart canes with laser SLAM and visual SLAM, capable of recognizing obstacles in front of the visually impaired, recognizing faces, and completing daily tasks, with tracking functions, have been developed and constantly updated. Each type of laser and vision-sensing smart cane has its own advantages and disadvantages.

Some smart cane systems have fewer obstacle-recognition functions, a lack of targeted recognition of obstacles on the road, and the system response speed is slow. In this regard, Slade P et al. (2021) [13] proposed an active smart cane based on laser SLAM technology, which was equipped with RPLIDAR-A1 LiDAR, a Raspberry PI monocular camera, IMU, and GPS. The main control module of the smart cane was Raspberry PI 4B, and the monocular camera used was the official camera of Raspberry PI, so the real-time target recognition ability and recognition efficiency of the smart cane were not high quality, and it recognized less objects. A Carranza et al. (2022) [14] proposed a smart guide system consisting of Raspberry PI 4, a Raspberry PI official camera, ultrasonic sensor, and speaker. However, the accuracy rate of the intelligent cane for vehicle recognition was 55–56%, the recognition types were few, and the recognition accuracy was low. TK Chuang et al. (2018) [15] proposed a guide dog smart cane with three Raspberry PI 2Bs, one NVIDIA Jetson TX1, and three cameras installed on the system. However, the system lacked the ability to identify other obstacles, and its ability to work was reduced when deviating from the preset trajectory. In terms of targeted obstacle recognition, Y Jin et al. (2015) [16] proposed a smart cane that could detect and recognize the faces of strangers around the visually impaired person. However, the smart cane system could only recognize faces at a limited range, and the number of faces that could be recognized was also very limited. K Jivrajani et al. (2022) [17] proposed an AIoT-based smart cane, which used Raspberry PI 3 as the main control module and OmniVision OV5647 as the camera module. The system could identify some daily necessities, but the system lacked the ability to identify obstacles on the road. To solve the problem of the slow system response, H Hakim et al. (2019) [18,19] proposed an indoor visually impaired assistance device based on the two-dimensional laser SLAM. But the identification process ran online in the cloud, and the detection frame rate was only 0.6 FPS, making the recognition rate of the system low. B Kumar (2021) [20] proposed a ViT Cane, a visual assistant for the visually impaired. Because the ViT Cane cannot recognize fast-moving vehicles on real roads, the system’s response rate was slow, so it was not able to be used on real roads. N Kumar et al. (2022) [21] proposed a smart guide system based on YOLOv5, which used Raspberry PI 3 as its main control. The time required for image data analysis and processing varied from less than 1 s to 3 s, so there was a certain delay in the process.

Some smart cane systems lack the functionality of actively guiding the visually impaired. In this regard, Z Xie et al. (2022) [22] proposed a multi-sensory navigation system that combined YOLO and ORB-SLAM. However, the smart cane lacked a driving device, the navigation ability of the motor array tactile feedback navigation was weak, and lacking the functionality of actively guiding the visually impaired. L Udayagini et al. (2023) [23] proposed a smart cane with Raspberry PI 3 as the main control board, combining HCSRCO4 ultrasonic sensor, Raspberry PI official camera module, and acceleration sensor and humidity sensor. The smart cane adopted yolov3 algorithm to recognize the faces of relatives of the visually impaired and surrounding vehicles and other obstacles. However, the design of the smart cane was too simple, with weak object recognition capability and no active navigation function. Cang Ye et al. (2014) [24] proposed a smart cane named CRC. The CRC smart cane was equipped with an active rolling tip with an encoder servo motor and electromagnetic clutch that can change direction, but the active rolling tip could only provide navigational steering, and lacked the function of actively guiding the visually impaired to avoid obstacles and navigate. S Agrawal et al. (2022) [25] proposed a visual SLAM smart cane with tactile navigation. However, the Visual SLAM smart cane lacked the ability to obtain other information in the environment, so the smart cane cannot not measure the distance such as to the obstacles ahead, without navigational guide functionality, and cannot, for example, actively guide the visually impaired person to an empty chair. H Zhang and C Ye et al. (2017–2018) [26,27] developed an indoor visual SLAM real-time path finding navigation system for the visually impaired based on an SR4000 3D-TOF camera, and proposed a new two-step pose image SLAM method. However, the smart cane guided users through voice interface navigation commands, so it lacked the functionality of actively guiding the visually impaired. H Zhang et al. (2019) [28] proposed a visual inertial SLAM navigation aid smart cane for the visually impaired combined with the RealSense R200 camera and the VectorNav VN100 IMU module, whose appearance was similar to that of [24]. Jin et al. (2009) [29] proposed the EYECane guide system, which notified the visually impaired through auditory information and enables them to passively listen to the path information planned by the system to walk. Based on Viola Jones and TensorFlow algorithm, U Masud et al. [30] proposed a smart cane composed of Raspberry PI 4B and Raspberry PI camera (V2) that can recognize faces and surrounding obstacles. However, the smart cane lacked a driving device, so it could only navigate passively. K Suresh (2022) et al. [31] proposed a smart cane with image recognition function with Arduino as the main control module. The smart cane used the YOLOv4 algorithm to identify people, mobile phones, and stairs, and the recognition accuracy rate was above 90%, so the movement speed of the visually impaired could be increased by 15% to 20%. However, the smart cane still lacked active guidance.

Some smart cane systems need to be assisted by electronic mobile devices [32,33] and an external computer as the main control [34,35,36,37]. The smart cane assisted by electronic mobile devices will lead to an increase in the manufacturing costs of the smart cane, with limited functions, which is not conducive to the in-depth development of the system. Smart cane external computers will increase the burden on the visually impaired person’s body and affect the walking of the visually impaired person.

In summary, the current laser and vision-sensing smart canes have the following problems: (1) there are few obstacle recognition types, and a lack of targeted identification of obstacles on the road, and the system response is slow; (2) there is a lack of active guidance for the visually impaired; (3) the visually impaired will need an external computer as the main control and need to be assisted by electronic mobile devices. In view of the above problems, our starting point is to increase the identification types of obstacles on the road, improve the real-time response speed of the system, improve the active guiding capability for the visually impaired without external devices, and develop our smart guide system by combining Cartographer and the improved YOLOv5 algorithm.

In this paper, we propose a smart cane guide system based on 2D LiDAR and RGB-D camera sensing. The blind guide system combines Cartographer and the improved YOLOv5 algorithm to realize navigation and target recognition functions. We have transformed the commonly used travel aids (the white cane) for the visually impaired, endowed them with intelligence, and equipped the intelligent guide system on the guide cane to better solve problems such as obstacle avoidance, navigation, and obstacle recognition encountered by the visually impaired in the process of walking. Our work can be summarized as follows:(1)The Cartographer algorithm was used to enable the smart cane to acquire laser SLAM function to achieve navigation capabilities, and the improved YOLOv5 algorithm was used to enable the smart cane to identify obstacles in front of the visually impaired person.(2)The smart cane system and structure were designed based on 2D LiDAR sensing and an RGB-D camera, and the intelligent blind guide system was equipped on the smart cane for actual functional testing.(3)The field test showed that the intelligent guide system was real-time and effective in helping the visually impaired navigate indoors and outdoors and to identify obstacles.

## 2. System Design

This work developed an intelligent guide system with navigation and object recognition functions to help the visually impaired navigate to a preset destination and identify obstacles on the road. The intelligent guide system is composed of four parts: a data acquisition module, a central processing module, a driving module, and a power supply module. The framework of the intelligent guide system is shown in Figure 1.

### 2.1. Smart Cane Structure Design

The smart cane in the literature [13] moves forward through the left and right swing of the omnidirectional wheel. Its way of moving forward is based on the process of the visually impaired person holding an ordinary blind cane swinging left and right and tapping the ground to detect the obstacles ahead. This method of moving forward is conducive to the quick mastery of this intelligent cane for the visually impaired who have been used to the ordinary cane; however, in the case of the smart cane, it already has a sensor to detect the road and obstacle information in front of it, so the efficiency of moving forward will be affected. In the literature, the smart cane robot [38,39] uses a rod similar to a white cane to connect to a robot car, and drives the rod through the movement of the robot car to assist the visually impaired person in walking. At this time, the robot car with two wheels is equivalent to acting as an “intelligent guide dog” to pull the visually impaired person in walking, but this way of moving will make it difficult for the visually impaired person to adapt to being pulled, as they are used to holding an ordinary white cane. Inspired by the literature [13,38,39], we combined the advantages of different studies and designed the laser and vision-sensing smart cane structure by using the omnidirectional wheel and two differential wheels to cooperate with each other.

In this paper, Soildworks-2021 3D CAD software is used to design the appearance structure of the laser and vision-sensing smart cane, and its appearance and design are shown in Figure 2. The smart cane adopts lightweight and easy to carry and replaceable components, mainly composed of data acquisition and processing devices (LiDAR, RGB-D camera, IMU, GPS), a white cane, bracket, omnidirectional wheel, coding motor, wheel, and a lithium battery. The omnidirectional wheel and two main wheels of the smart cane are in contact with the ground and jointly support the whole smart cane, which can ensure that the smart cane can be in a stable state without interference from external forces, and the omnidirectional wheel and main wheels give the smart cane active navigational ability. In addition, the height and center of gravity of the smart cane can be adjusted according to the height of the visually impaired, so that he/she can use the smart cane more easily.

### 2.2. Smart Cane System Hardware

The hardware design of the smart cane system includes: main control module: NVIDIA Jetson Nano B01 (4 GB); data acquisition module: M10P TOF 2D LiDAR, Gemini binocular depth camera, N100N 9-axis inertial navigation module, G60 GPS Beidou dual-mode positioning module; drive module: STM32F407VET6 microcontroller, 500-wire AB phase GMR (Giant Magnetoresistance) encoder, double bearing 75mm diameter omnidirectional wheel, two 85 mm diameter non-slip rubber tires; power module: 12V-9800MAH lithium battery. The hardware parameters of the intelligent guide system are shown in Table 1. The total weight of the smart cane is 2.65 KG, and the total cost of manufacturing the smart cane is CNY 5432.

The central processing module, Jetson Nano B01 (4 GB), can receive the environmental information data of the data acquisition module and analyze and process the data. After data analysis and processing, the instructions are transmitted to the driving module STM32F407VET6, so as to drive the smart cane when walking. The LiDAR can obtain the position and distance information of the obstacles ahead, achieve synchronous positioning and composition (SLAM) function, and enable the smart cane to avoid obstacles and go to the destination according to the small-range planned route. The depth camera can identify what object is the obstacle in front of the smart cane, identify the tactile paving, pedestrian crossing, traffic lights, warning posts, and stone pillars around the visually impaired person, and can detect the ground potholes, steps, and low height obstacles that the LiDAR cannot detect. The inertial measurement unit (IMU) is composed of a three-axis gyroscope, accelerometer, and magnetometer. The IMU module can detect the attitude information during the movement of the smart cane and prevent deviations caused by the accumulated error during the movement of the smart cane. The GPS module can detect the geographical location of the smart cane in real time, provide a large range and long distance navigation information for the smart cane, and record its moving trajectory. The microcontroller STM32F407VETb is used to accept the instructions transmitted by the central processing module and drive the GMR high-precision coding motor to move the non-slip wheel and the omnidirectional wheel. The physical appearance structure of the laser and vision-sensing smart cane is shown in Figure 3, and the hardware structure of the intelligent blind guide system is shown in Figure 4.

### 2.3. Working Process of the Intelligent Guide System

The working process of the smart cane is as follows: When the visually impaired person uses the smart cane, he/she turns on the power switch of the smart cane to supply power to each module of the smart cane, so that it can enter the working state. The visually impaired person can input the destination to the smart cane system through the voice module; the central processing module starts the data acquisition module to collect the obstacle distance and image data of the environment around the visually impaired person. After data processing and analysis, the central processing module transmits the instructions to the drive module, so that the smart cane can realize positioning, obstacle avoidance, navigation, attitude detection, and other functions in real time both indoors and outdoors, and identify the obstacles ahead. The visually impaired person is guided by the smart cane to move around the pedestrians and obstacles on the road and reach the destination. At the same time, GPS will record the time node and travel trajectory of the visually impaired when they go out. When he/she stops using the smart cane, the charger can be plugged into the smart cane to charge the battery.

## 3. Materials and Methods

### 3.1. Cartographer Algorithm

The commonly used LiDAR-based simultaneous localization and mapping (SLAM) technology can accurately measure the distance and angle of the obstacle point of the visually impaired and generate an environment map that is convenient for the navigation of the intelligent guide system, which has become an indispensable technology in the field of blindness guide equipment. The two-dimensional laser SLAM scheme currently applied in the field of blindness includes Gmapping-based particle filter [40], Hector SLAM based on scan-to-scan [41], and Cartographer based on pose optimization [42]. Compared with other 2D laser SLAM schemes, Cartographer provides accurate solutions for positioning and map construction, using global map optimization cycles and local probabilistic map updates. This makes the application of Cartographer’s laser SLAM system more robust to environmental changes [43]. Therefore, in this paper, we used Cartographer algorithm to realize the positioning, obstacle avoidance, and navigation functions of the intelligent guide system.

Cartographer [44] is a 2D laser SLAM system with scan-to-submap matching with loop detection and graph optimization developed by Google Inc. (Mountain View, CA, USA). The Cartographer algorithm consists of two separate but related subsystems, local SLAM (front-end matching) and global SLAM (back-end optimization), both of which have the function of optimizing the pose. The map of Cartographer is composed of submaps. Submaps are constructed by constantly calibrating the scan point set and submap coordinates; that is, scan is converted to the submap coordinate system. Local SLAM builds and maintains a series of submaps through a series of continuous laser scans, which are optimally inserted into the submap using Ceres scan matching when a new laser scan is performed. However, submaps create cumulative errors. Global SLAM uses loop closure to eliminate the accumulated errors in local SLAM and correct the pose between each submap. In order to achieve real-time loop closure, a branch-and-bound method is used to speed up the search process while reducing the amount of computation to calculate the scan-to-submap match as a constraint. The Cartographer algorithm mainly reduces the computational complexity of loopback detection to improve back-end optimization efficiency. The Cartographer algorithm architecture is shown in Figure 5.

### 3.2. Improved Yolov5 Algorithm

YOLO (You Only Look Once) [45,46,47] is an object detection algorithm. YOLOv5 [48] was proposed by Ultralytics LLC (Washington, DC, USA), which has been widely used in the assistance system for the visually impaired with target recognition functions. It is used to identify objects such as pedestrian crossings [49,50], traffic lights [51], buses [52], straight or winding paths [21], clothing defects [53], stairs and roads [54], faces and money [55], and indoor fires [56]. Since the official model of YOLOv5 alone cannot meet the requirements of this work to identify all obstacles on the road and improve the training speed of the YOLOv5 model, in this paper, we increased the training set of the guide system model and added the attention machine [57] system to the YOLOv5 algorithm to make improvements. We replaced the C3 layer in the backbone of the YOLOv5 algorithm with the convolutional block attention module (CBAM) [58]. The CBAM is a lightweight attention module, the CBAM attention mechanism is composed of a channel attention module (CAM) and a spatial attention module (SAM). The channel attention module and the spatial attention module focus on the “what” and “where” in the input image, respectively. The channel attention module deals with the distribution relationship of feature graph channels by focusing on the “what” of the input image to strengthen the weight of a channel. By focusing on the “where” of the input image, the spatial attention module can make the neural network pay more attention to the pixel region that plays a decisive role in image classification and ignore other unimportant pixel regions. The CBAM attention mechanism allocates attention to both channel and space, which enhances the effect of the attention mechanism on the overall model performance. The improved YOLOv5 algorithm architecture is shown in Figure 6.

## 4. Experiment and Results

### 4.1. Simulation Experiment

The computer processor used in the simulation experiment of this work was the Intel(R) Core(TM)i5-11300H. We installed the ROS Noetic open source robot operating system under Ubuntu 20.04 Linux64-bit operating system on virtualbox 6.1.16 open source virtual Machine software in the calculator processor.

In the ROS Noetic simulation experiment, a URDF file was used to create a simulated intelligent cane model, and a Xacro (XML Macros) file was used to optimize the macro encapsulation of the URDF file, so as to optimize its code structure and improve its code reuse rate. Then, the motor control board Arbotix function was invoked to realize the movement of the simulated smart cane model in Gazebo and RVIZ in ROS Noetic. The smart cane model was composed of two environmental information sensors, LiDAR and RGB-D camera, blind cane, data acquisition and processing device, support, all-directional wheel, and two main wheels. The simulated smart cane model was similarly designed via reference to the appearance design of the smart cane in Section 2.1 of this paper. The simulated URDF model design drawing and the side view of all sides are shown in Figure 7, and the simulated URDF model hierarchical relationship is shown in Figure 8.

To simulate laser SLAM, we installed Gazebo (3D Dynamic simulator) and RVIZ (Robot Visualization—an open source tool for visualizing robot systems) in ROS Noetic under Ubuntu 20.04 Linux 64-bit operating systems. First, we created a simulation environment of the smart cane system in Gazebo to accurately and effectively simulate the smart cane in the complex indoor environment for map construction and navigation. We added several simulated obstacle models with different shapes to the Gazebo simulation environment, including four walls, three cylindrical obstacles, and three cuboid obstacles. Additionally, the smart cane URDF model was displayed in Gazebo. Then, we displayed the simulation environment and URDF model of the simulated smart cane in RVIZ. RVIZ can receive and display real-time data from sensors on the smart cane (such as the LiDAR and RGB-D camera), and obtain point cloud data around the simulated smart cane through LiDAR. The RGB-D camera was used to obtain the simulation environment picture information data in front of the smart cane, in order to help the smart cane to know the surrounding environment information. In RVIZ, we set the navigation target point of the smart cane through 2D Nav Goal, and the URDF model of the simulated smart cane built a two-dimensional map of the simulation environment through the LiDAR sensor, and the final map was almost consistent with the simulation environment. The Cartographer algorithm was used to plan the navigation route from the starting point to the target point. Finally, the simulated smart cane achieved the function of simulation laser SLAM by avoiding obstacles and following the path planned by the system (green dotted line) to the preset navigation destination, and the simulation laser SLAM rate was 26–27 FPS. The simulation of the smart cane to achieve laser SLAM function is shown in Figure 9.

### 4.2. Laser SLAM Experiment

In our laser SLAM experiment, the Ubuntu 18.04 LTS64-bit operating system was deployed on Jetson nano B01, the main control module of the smart cane, and then Melodic-ROS was installed on the Ubuntu system. Additionally, the Jetson nano B01 was externally connected to the M10P TOF 2D Lidar. We used the Cartographer algorithm on Melodic-ROS to realize the laser SLAM function, and set target points through 2D Nav Goal in RVIZ in Melodic-ROS, so that the smart cane could sense and avoid obstacles, and plan the route to the preset destination.

We simulated the scene of a visually impaired person walking in the corridors and floor passages of the teaching buildings in the Longkun South campus, Hainan Normal University. The testers closed their eyes and held the smart cane in their hands to simulate the field test of a visually impaired person using the smart cane. The laser SLAM field test of the smart cane is shown in Figure 10. The smart cane system builds the map and realizes the navigation function of the surrounding environment on the corridor and floor passage of the teaching building. In the field test, the mapping and positioning accuracy of the laser SLAM of the smart cane system was 1 m ± 7 cm, and the laser SLAM rate of the smart cane system was 25~31 FPS. Although there is some delay in the process of field mapping and navigation, it does not affect the real-time mapping of the surrounding environment. The test results showed that the smart cane system can indeed realize laser SLAM using the Cartographer algorithm on 2D LiDAR; thus, it realized the map construction of the environment around the visually impaired person and realized a short-distance obstacle avoidance navigation.

### 4.3. The Improved YOLOv5 Algorithm Realizing Obstacle Detection

In this work, an improved YOLOv5 algorithm was used to achieve obstacle detection. The computer processor used in the training experiment of the improved YOLOv5 obstacle detection model was the Intel(R) Core(TM)i5-11400F, and the GPU graphics card was NVDIA GeForce RTX 2060. We conducted YOLOv5 obstacle detection model training on the collected and labeled obstacle data set in cuda 11, the miniconda 3 environment management tool, python 3.10, pytorch 2.0.1, and other environments. We added the convolutional block attention module (CBAM) to the official code of the YOLOv5 algorithm V6.0. CBAM is a lightweight and general feedforward convolutional neural network attention module. It focuses on the target object information that is more critical to the current task, reduces attention to other non-target object information, and can even filter out irrelevant information to improve the efficiency and accuracy of task processing, so as to improve the overall performance of obstacle recognition model.

First, 104 videos were collected on multiple sections of the Longkun South Road, Qiongshan District, Haikou City, Hainan Province, and 5337 pictures containing effective obstacle information were captured from the collected videos (including 4137 in the training set and 1200 in the test set). LabelImg software (version: 1.8.6) was used to label all the pictures in the obstacle training set. A total of 13,193 annotation boxes (six types) of self-made data sets “traffic lights (green light state), traffic lights (red light state), pedestrian crossings, warning columns, stone pillars, tactile paving” were labeled. Our overall guide data set used the COCO 2017 official data set (80 categories: pedestrians, vehicles, etc.) plus a self-made obstacle data set (6 categories), with a total of 86 categories of objects and 121,308 images. The total data set of this intelligent guide system is shown in Figure 11.

Then, the total data set of the intelligent guide system was trained with the improved YOLOv5 algorithm. The initial training design was 300 training rounds (epoch = 300), and each time, four pictures (epoch size = 4) were input into the neural network. The final result was a total of 300 rounds of training (epoch = 300), each training round lasted about 35 min and 30 s, and the total training time was 182.709 h. The improved YOLOv5 obstacle recognition model of the guide system that we needed was obtained through training. The obstacle recognition model results trained by the improved YOLOv5 are shown in Figure 12. Therefore, the obstacle model trained in this work can identify 86 types of objects. It can effectively recognize the target of pedestrians, motorcycles, cars, warning posts, stone piers, pedestrian crossings, traffic lights (green and red), and tactile paving, among others. According to the analysis of the trained improved YOLOv5 model, the recognition rate of the improved YOLOv5 model for pedestrian crossing was 84.6% and the recognition rate for vehicles was greater than 71.8%. The overall recognition rate of the system for 86 types of objects was 61.2%. The recognition rate, recall rate, mAP, and mAP50-95 of obstacle targets on some roads of the improved YOLOv5 model are shown in Table 2. At the same time, we used the same data set to train the unimproved YOLOv5 model, and the overall recognition rate of the unimproved YOLOv5 model was about 56.4%. After comparative analysis, the overall recognition rate of the improved YOLOv5 model (YOLOV5-CBAM) was about 4.8% higher than that of the unimproved YOLOv5 model.

Next, we conducted training on the improved YOLOv5 obstacle recognition model on the computer processor Intel(R) Core(TM)i5-11400F, the GPU graphics card was the NVDIA GeForce RTX 2060. The inference was carried out in cuda 11, the miniconda 3 environmental management tool, python 3.10, pytorch 2.0.1, and other environments, and the inference results showed that the intelligent guide system could effectively identify objects such as pedestrians, vehicles, tactile paving, pedestrian crossings, and traffic lights on the road. The reasoning results of the improved YOLOv5 obstacle recognition model are shown in Figure 13.

Next, we deployed the trained and improved YOLOv5 obstacle recognition model to the Jetson nano B01 system. We installed the Ubuntu18.04 Linux 64-bit operating system on Jetson nano B01 and performed environmental configuration. The environmental versions deployed by the Jetson nano B01 system are shown in Table 3. Then, the improved YOLOv5 obstacle recognition model deployed on Jetson nano B01 was converted into the TensorRT format to optimize the obstacle recognition model and improve the speed of the obstacle recognition model running on the Jetson nano B01 GPU.

Finally, we attached an ORBBEC Gemini RGB-D camera to the Jetson nano B01, and mounted the guide system on the smart cane. We carried the smart cane to the entrance of the library of Hainan Normal University and carried out field tests on several sections of Longkun South Road in Haikou City, Hainan Province, to simulate the process of a visually impaired person identifying obstacles in front of them during walking. The tests of the improved YOLOv5 obstacle recognition on campus and on the road are shown in Figure 14 and Figure 15.

The test results show that: (1) The intelligent guide system that we developed can effectively identify the objects in front of the visually impaired person, such as pedestrians, cars, motorcycles, trucks, pedestrian crossings, and traffic lights. The recognition rate of the intelligent guide system for a pedestrian crossing was 84.6%, that for vehicles was more than 71.8%, and the overall recognition rate of the system for 86 types of objects was 61.2%; (2) The number of frames transmitted per second for target recognition was 25 to 26 FPS, so the visually impaired person can completely identify obstacles in real time without much error; (3) In the field test process, under the following conditions, (a) with interference of strong sunlight during the day, as shown in Figure 16, or (b) without enough light in the surroundings at night, as shown in Figure 17, or (c) with cars moving too fast on the road, etc., the obstacle recognition rate will be reduced; however, to some degree, obstacles in front of the visually impaired can still be identified.

## 5. Discussion

Compared with many other intelligent guide systems, the main advantage of the intelligent guide system proposed in this work is that the distance between the smart cane and the obstacle can be measured using two-dimensional LiDAR to achieve laser SLAM, and at the same time, it can identify obstacles in front of the visually impaired person, improve the real-time response speed of the intelligent guide system, increase the types of obstacle recognition, and expand the overall detection range. We performed experimental simulations for the process of a visually impaired person leaving the tactile paving to the pedestrian crossing, waiting for the traffic light, avoiding obstacles, and walking on the campus road. The smart cane does not need to lay new tactile paving indoors and outdoors and specific electronic position information transceiver devices, and can actively provide direction guidance for the visually impaired in unfamiliar environments, avoid obstacles in front of them, guide them to the destination, and provide safety and convenience for the visually impaired.

### 5.1. The Choice of Main Control Module of Smart Cane

The laser and vision-sensing smart cane system originally used the smaller Raspberry PI 4B as the main control module, but the research and development process summary found that the Raspberry PI 4B has a slow response speed and weak real-time processing ability. Because Raspberry PI 4B lacks a complete GPU and therefore lacks deep learning capability, there is a relatively large delay when using Raspberry PI 4B for YOLOv5 target recognition.

We made some improvements to the recognition rate of the Raspberry PI 4B, which are as follows:

We used the Raspberry PI 4B as the video image transfer server. The images obtained by the RGB-D camera do not use the CPU of the Raspberry PI 4B for reasoning, but use the rented Ali Cloud server Ubuntu20.04 (2 core CPU, 2 GB memory, 60 GB ESSD, 5 Mbps) for video image transfer. A real time messaging protocol (RTMP) stream server was built using nginx server software, and ffmpeg was used to push the stream. The video images were pushed to Intel(R) Core(TM)i5-11400F. The GPU graphics card is the calculator processor of NVDIA GeForce RTX 2060 for inference recognition. Though we made such improvements, the recognition rate of the improved Raspberry PI 4B still has a delay of more than 10 s, and cannot recognize the obstacles in front of the smart cane in real time.

Then, in this work, we compared and analyzed the obstacle recognition performance of Jetson Nano B01 and Raspberry PI 4B. In the case of the same YOLOv5 obstacle model, the speed of object recognition of Raspberry PI 4B is only about 0.444 FPS. The Jetson Nano B01 can recognize objects at up to 26 FPS. It can be seen that the object recognition rate of Jetson Nano B01 in this work is much higher than that of Raspberry PI 4B, and the recognition rate of Jetson Nano B01 is about 58.6 times that of Raspberry PI 4B. The comparative analysis of object recognition rates is shown in Figure 18. Because Jetson Nano B01 has a complete GPU and a fast response speed, Jetson Nona B01 is superior to Raspberry PI 4B in this work due to its data processing, image acquisition, recognition rate, and other performance. Therefore, we chose the Jetson Nona B01(4 GB) with deep learning capability as the main control module of the smart cane.

### 5.2. Limitations of This Work

There are still some limitations in this work, and more efforts should be made to improve. On the one hand, the laser SLAM mentioned in this work uses two-dimensional LiDAR, so the guide system can only obtain the obstacle distance information at the same height as the two-dimensional LiDAR on the smart cane. Such distance information can only know the local distance information of the obstacles in front of the visually impaired person, but cannot obtain the overall distance information of the obstacles in front of the visually impaired person. As a result, there are obstacles that cannot be measured at a lower height and obstacles that are suspended in mid-air. On the other hand, although the YOLOv5 model of the smart cane can identify 86 types of objects, the 86 types of objects do not contain sunken ground obstacles such as potholes and downward steps. When the visually impaired person encounters unavoidable obstacles sunk into the ground in the process of walking, such as potholes and downward steps, there is a risk of not being able to avoid those obstacles in time, which may harm the visually impaired person.

### 5.3. Future Work

#### 5.3.1. Using 3D LiDAR

Since the two-dimensional Lidar cannot measure the obstacles at a low height and obstacles hanging in the air, in our following work, we consider replacing the two-dimensional Lidar of the smart cane with the three-dimensional 16-line TOF LiDAR, so as to obtain the three-dimensional point cloud information of the obstacles in front of the visually impaired person and to realize the Cartographer 3D SLAM. Additionally, we will install ultrasonic modules on the bottom and top of the smart cane to measure the distance of obstacles that are low down in front of the smart cane or suspended in mid-air.

#### 5.3.2. Increasing the Number of Obstacles to Be Identified

Since the camera could not identify the potholes on the road, descending steps, and other obstacles sunk into the ground, we considered collecting enough picture information about potholes and descending steps on the road in the next work, and adding these pictures to the process of training the YOLOv5 obstacle detection model for marking and training. In this way, the YOLOv5 obstacle detection model that can identify more obstacles can be trained, so that the smart cane will have the ability to identify obstacles sunk into the ground.

#### 5.3.3. Fusion of LiDAR Point Cloud and Camera Image Data

In the future work, we will conduct the joint calibration of the camera and Lidar on the smart cane, and integrate and complement the camera image information and LiDAR point cloud data. Fusion of the image and point cloud data can not only overcome the shortcomings of a single sensor in environmental perception, but also obtain more abundant target observation data and improve the environmental perception accuracy of the smart cane.

## 6. Conclusions

We proposed an intelligent guide system based on 2D LiDAR and RGB-D camera sensing, designed the appearance structure of the smart cane, and installed the developed intelligent guide system on the smart cane. The guide system on the smart cane uses two-dimensional Lidar, depth camera, IMU, GPS, Jetson nano, STM32, and other hardware to build a two-dimensional map of the visually impaired person’s surroundings, plans the navigation path based on the Cartographer algorithm, and uses the improved YOLOv5 algorithm to quickly and effectively identify the common obstacles on the road surface. The experimental results show that the mapping and positioning accuracy of the system’s laser SLAM is 1 m ± 7 cm, and the laser SLAM speed of this system is 25~31 FPS, which can realize the short-distance obstacle avoidance and navigation function both in indoor and outdoor envionments. It can effectively identify obstacles such as pedestrians, vehicles, tactile paving, pedestrian crossings, traffic lights (red light state, green light state), warning columns, and stone piers on the highway. The intelligent guide system can recognize 86 types of objects, among which, the recognition rate of the system is 84.6% for pedestrian crossings and more than 71.8% for vehicles. The overall recognition rate of the system is 61.2% for 86 types of objects, and the recognition rate of the intelligent guide system is 25~26 FPS. In summary, the intelligent guide system developed by us can effectively guide the visually impaired to the predicted destination, and can quickly and effectively identify the obstacles on the way.

## Figures and Tables

**Figure 1 sensors-24-00870-f001:**
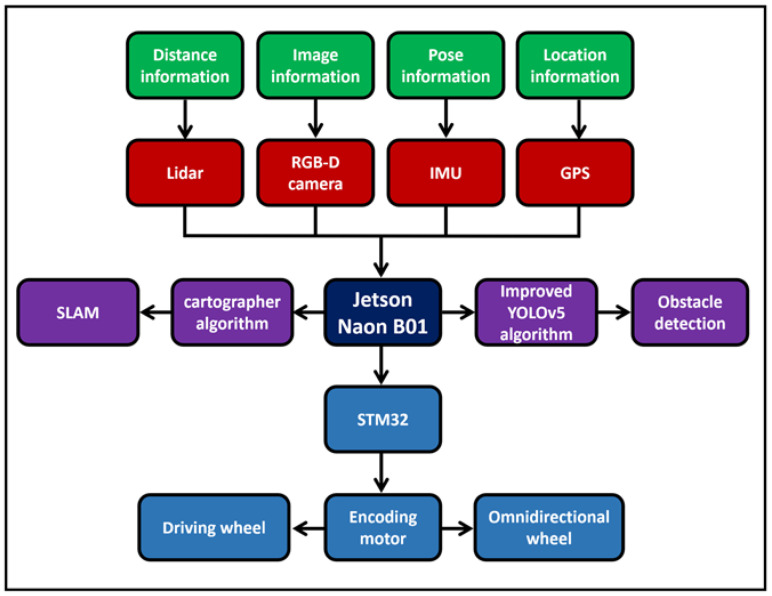
Intelligent guide system frame diagram.

**Figure 2 sensors-24-00870-f002:**
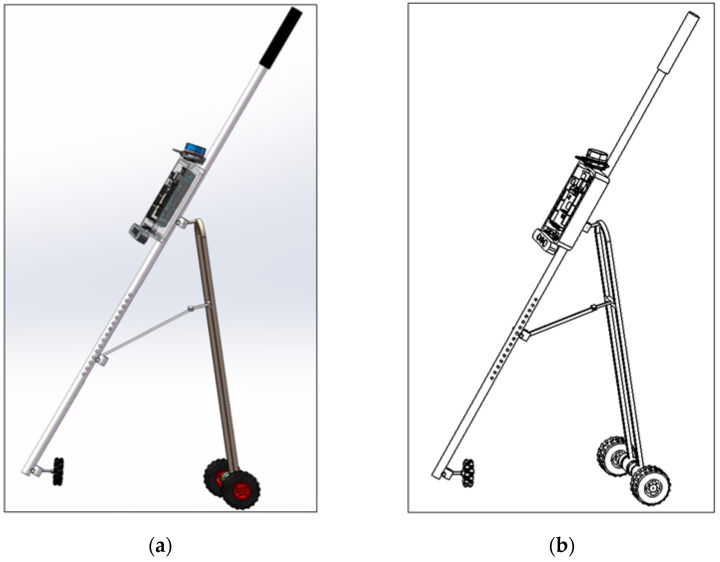
Laser and vision-sensing smart cane structure appearance design: (**a**) smart cane 3D solid model; (**b**) smart cane CAD diagram.

**Figure 3 sensors-24-00870-f003:**
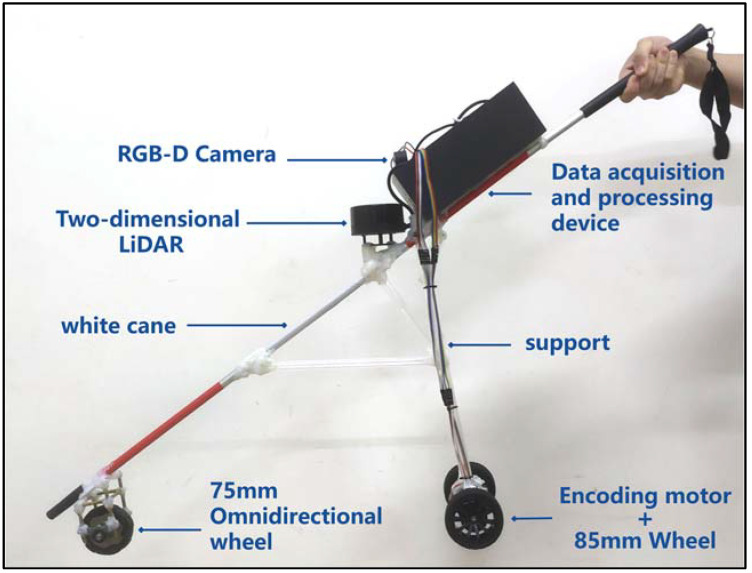
The physical appearance structure of the laser and vision-sensing smart cane.

**Figure 4 sensors-24-00870-f004:**
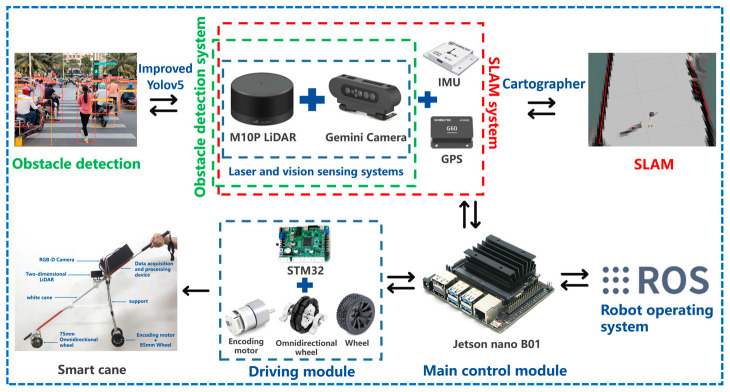
Hardware structure of intelligent guide system.

**Figure 5 sensors-24-00870-f005:**
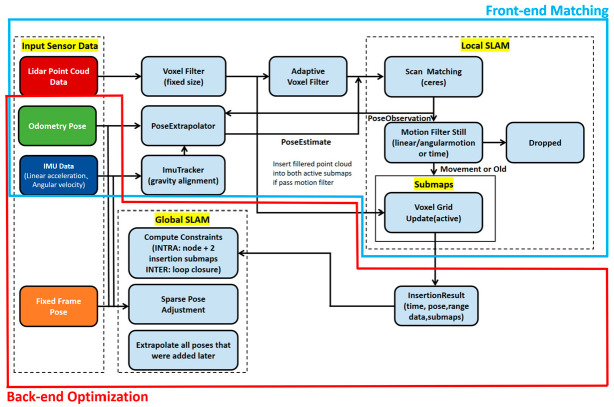
Architecture of two-dimensional laser SLAM Cartographer algorithm.

**Figure 6 sensors-24-00870-f006:**
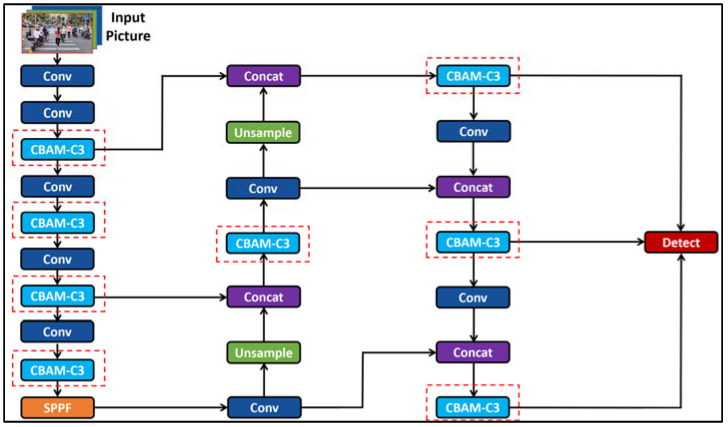
Improved YOLOv5 algorithm architecture (C3 layer replaced with CBAM).

**Figure 7 sensors-24-00870-f007:**
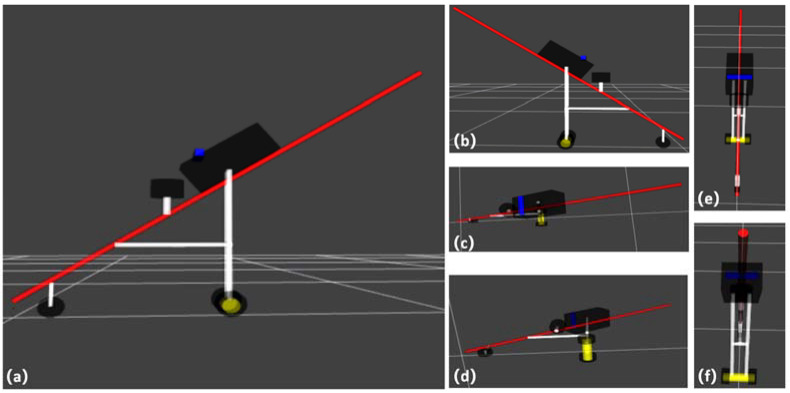
Simulated URDF model of smart cane: (**a**) main view; (**b**) rear view; (**c**) vertical view; (**d**) bottom view; (**e**) left view; (**f**) right view.

**Figure 8 sensors-24-00870-f008:**
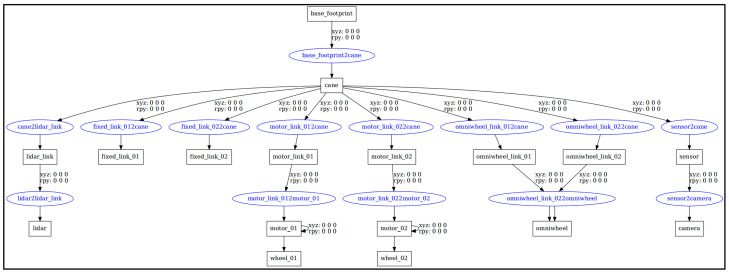
The simulated URDF hierarchical diagram of smart cane.

**Figure 9 sensors-24-00870-f009:**
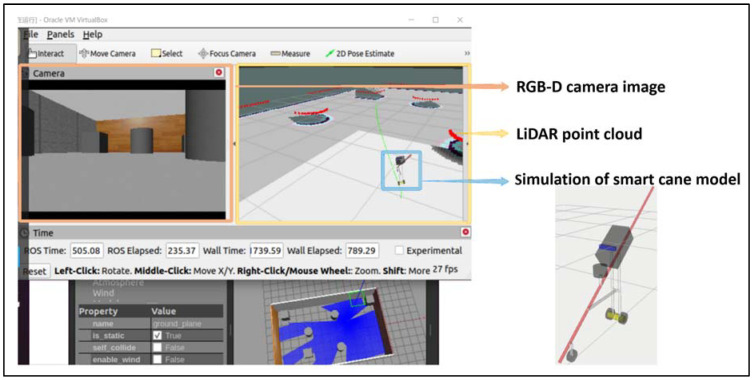
Simulation of smart cane to achieve laser SLAM (green dotted line is the path of system planning).

**Figure 10 sensors-24-00870-f010:**
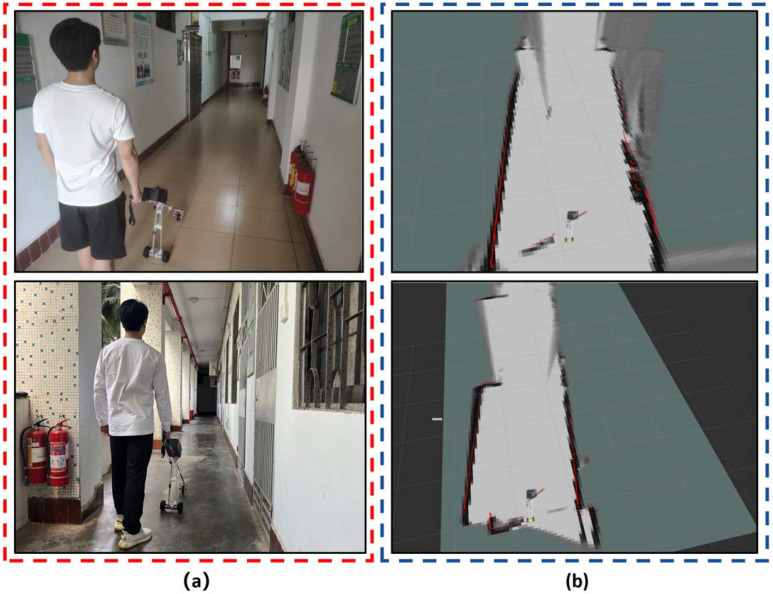
The smart cane uses the Cartographer algorithm to implement laser SLAM: (**a**) experimental site (corridor and floor passages); (**b**) smart cane achieves laser SLAM.

**Figure 11 sensors-24-00870-f011:**
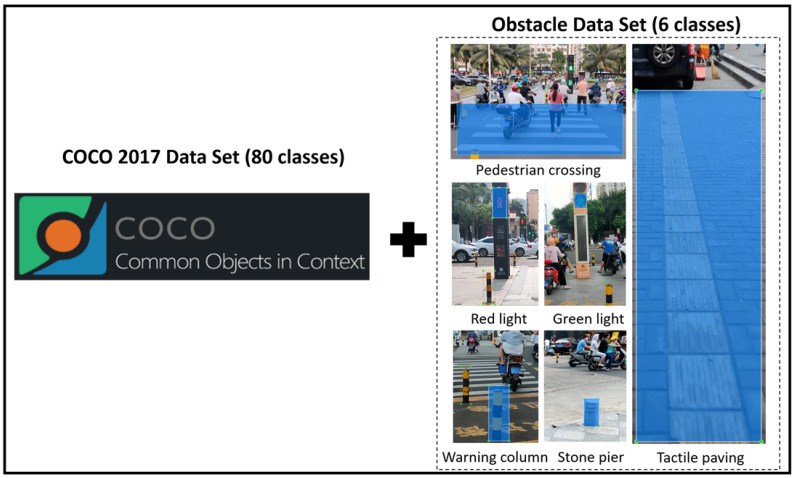
Total data set of improved YOLOv5 for intelligent guide system (COCO 2017 Data Set + Obstacle Data Set).

**Figure 12 sensors-24-00870-f012:**
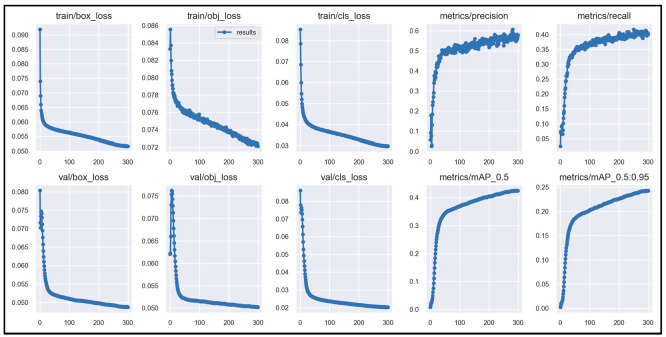
Results of obstacle recognition model trained by improved YOLOv5.

**Figure 13 sensors-24-00870-f013:**
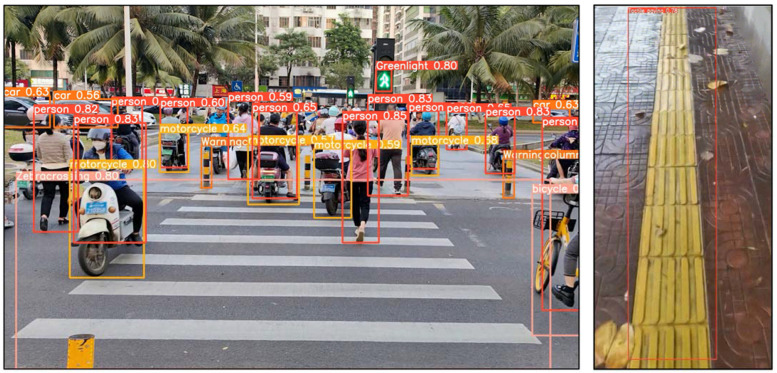
The improved YOLOv5 model identifies obstacles such as pedestrians, vehicles, tactile paving, pedestrian crossings, and traffic lights.

**Figure 14 sensors-24-00870-f014:**
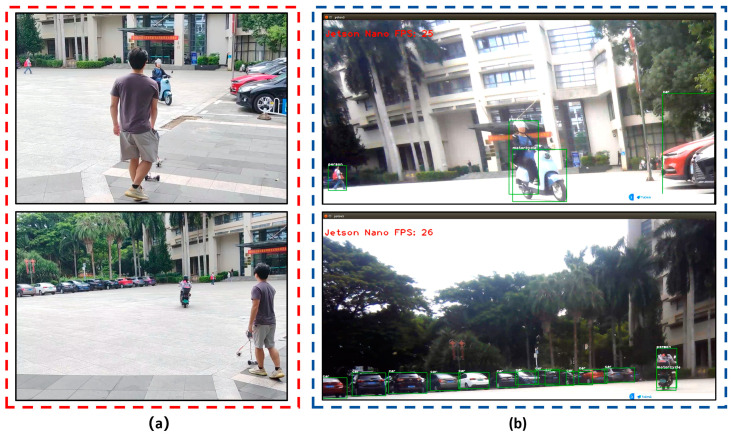
Improved YOLOv5 obstacle recognition for field tests on campus: (**a**) field test map on campus; (**b**) the improved YOLOv5 obstacle recognition can identify objects such as pedestrians, motorcycles, and cars.

**Figure 15 sensors-24-00870-f015:**
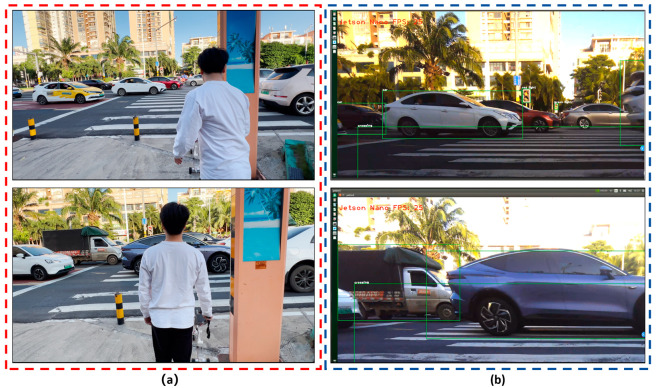
Field testing of improved YOLOv5 obstacle recognition on the road: (**a**) field test on the road; (**b**) improved YOLOv5 obstacle recognition, which can recognize objects such as cars, trucks, pedestrian crossings, and traffic lights (red light and green light).

**Figure 16 sensors-24-00870-f016:**
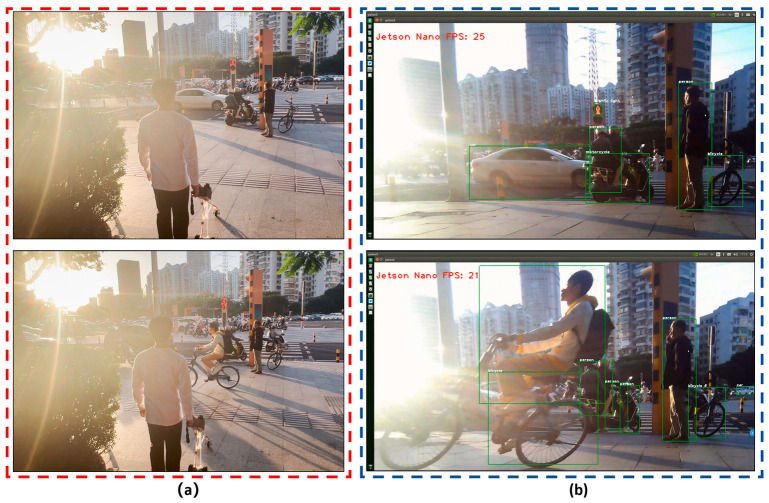
In circumstances of strong sunlight during the day, the improved YOLOv5 obstacle recognition was field-tested on the road: (**a**) a field test on the road in the face of strong sunlight; (**b**) the improved YOLOv5 obstacle recognition can identify people, motorcycles, cars, bicycles, traffic lights, and other objects.

**Figure 17 sensors-24-00870-f017:**
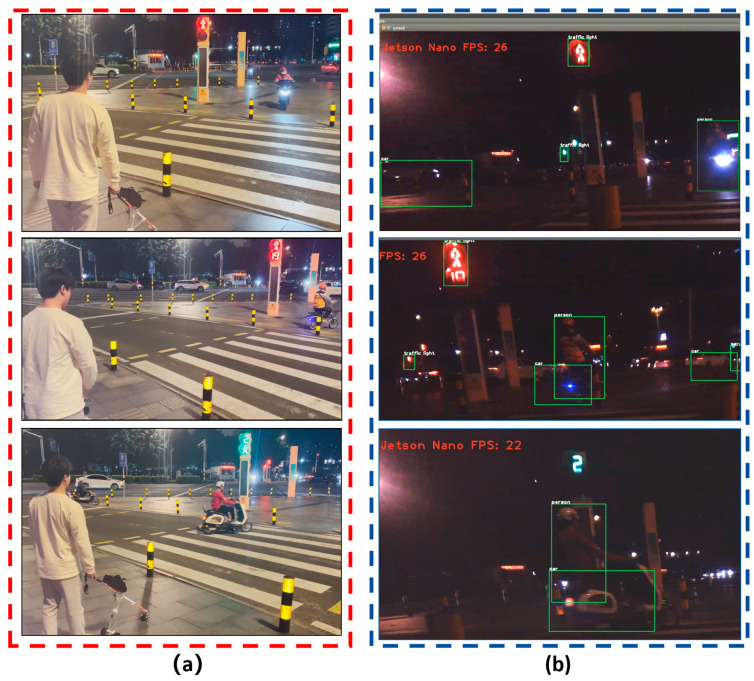
Field test of improved YOLOv5 obstacle recognition on the road in the absence of light in the surrounding environment at night: (**a**) field tests on the road at night when the surrounding environment lacks light; (**b**) the improved YOLOv5 obstacle recognition can identify objects such as people, cars, and traffic lights.

**Figure 18 sensors-24-00870-f018:**
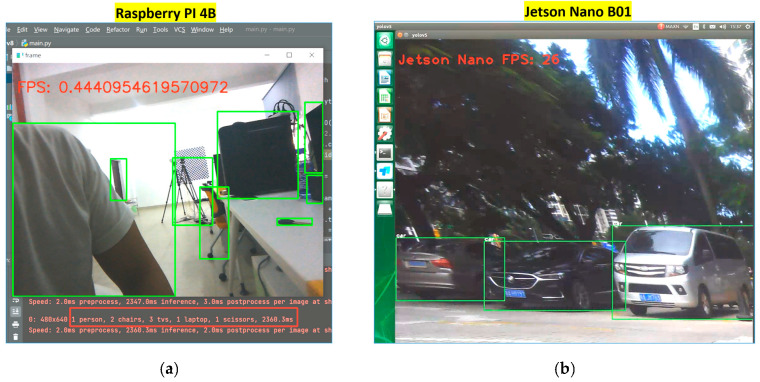
Comparison of obstacle detection speed between Raspberry PI 4B and Jetson Nano B01: (**a**) the Raspberry PI 4B obstacle detection at approximately 0.444 FPS; (**b**) Jetson Nano B01 obstacle detection speed: 26 FPS.

**Table 1 sensors-24-00870-t001:** Hardware parameters of intelligent guide system.

Hardware	Hardware Type	Parameters and Dimensions
Main control module	Jetson Nano B01(4 GB)	CPU:ARM Cortex-A57 GPU:128-core Maxwell
2D LiDAR	Leishen Intelligent System M10P TOF	Detection distance radius: 0–25 m Measurement accuracy: 1 m ± 3 cm Detection Angle: 360° Scanning frequency: 10 HZ
RGB-D camera	ORBBEC Gemini Pro	Detection accuracy: 1 m ± 5 mm Field of view(FOV): H67.9° × V45.3° Resolution @ Frame rate: 1920 × 1080 @ 30 fps
IMU	WHEELTEC 100 N	Static accuracy: 0.05° RMS Dynamic accuracy: 0.1° RMS
PGS	WHEELTEC G60	Positioning accuracy: 2.5 m
Microcontroller	STM32	STM32F407VET6
Encoding motor	WHEELTEC MG513	500 line, AB phase GMR Gear ratio: 1:30
Omnidirectional wheel	WHEELTEC omni wheel	Diameter: 75 mm Width: 25 mm
Wheel	WHEELTEC 85 mm	Diameter: 85 mm Width: 33.4 mm Coupling aperture: 6 mm
Battery	WHEELTEC 12V-9800MAH	Size: 98.6 × 64 × 29 mm^3^
White cane	j&x white cane	Length: 116 cm Diameter: 1.5 cm Weight: 0.23 kg

**Table 2 sensors-24-00870-t002:** The recognition rate, recall rate, mAP, and mAP50-95 of obstacle targets on the road of the improved YOLOv5.

Class	Precision	Recall	mAP50	mAP50-95
Person	0.667	0.634	0.67	0.388
Car	0.718	0.685	0.738	0.259
Motorcycle	0.61	0.447	0.505	0.473
Bus	0.72	0.625	0.666	0.488
Truck	0.798	0.666	0.75	0.481
Bicycle	0.543	0.389	0.392	0.183
Traffic light	0.583	0.361	0.374	0.176
Green light state	0.608	0.619	0.572	0.164
Red light state	0.616	0.516	0.572	0.354
Crossing	0.846	0.644	0.827	0.357
Warning column	0.783	0.856	0.724	0.51
Stone pier	0.692	0.633	0.669	0.42
Tactile paving	0.74	0.807	0.839	0.533

**Table 3 sensors-24-00870-t003:** Jetson nano B01 system deployment environment version parameters.

Environment	Version
Ubunut	18.04
Python	3.6.9
Pytorch	1.10.0
Cuda	10.2.300
CuDNN	8.2.1.8
Opencv	4.1.1
TensorRT	8.2.1
Jetpack	4.6.1
Machine	aarch64

## Data Availability

Data are contained within the article.

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
