# Peer review of "A Smart Cane Based on 2D LiDAR and RGB-D Camera Sensor-Realizing Navigation and Obstacle Recognition"

_sensors, 2024, doi:10.3390/s24030870_

Round 1
Reviewer 1 Report
Comments and Suggestions for Authors
(1) Whether the improved algorithm(YOLO 5) has been compared with other algorithms to verify the effectiveness and superiority of the improved algorithm? It is recommended to include comparative analysis in the paper to evaluate the performance and effectiveness of the algorithm comprehensively.
(2)What is the ability of the smart guide cane to identify sunken ground obstacles, which is crucial for guiding the blind ?
(3) How to integrate Camera image and LiDAR point clouds to balance the recognition accuracy and processing speed when identifying obstacles?
(4) Is the angle of the guide cane sensitive to image acquisition and will it affect the recognition effect?
Reviewer 2 Report
Comments and Suggestions for Authors
1. Please center your figures (e.g. Fig 1).
2. Turning the smart cane on and off: how is the blind impaired aware of the on/off status of the smart cane? is there any feedback (e.g. tactile) so they can have some sense of the trajectory planned by the cane?
3. What is the total weight of the cane with all the HW components? what is the weight of a standard (not smart) cane?
4. What is the cost of HW components and the modified cane compared to a standard one?
5. The Cartographer algorithm should be described in more details. What are the subgraphs the authors refer to? what is the local/global subsystem? what is the input and what is the output?
6. The sentence "In order to generate better subgraphs, the front-end architecture is designed to generate better subgraphs" does not convey real information.
7. The use of YOLOv5 is unclear. Have the authors trained the algorithm? how many images/objects were used to train the algorithm? what are the CAM/CBAM/SAM modules mentioned by the authors? what about more recent versions of YOLO (e.g. YOLOv8)?
8. What is "distribution relationship of feature graph channels"?
9. 4.1: The description of the simulation environment is lacking details. What objects were placed in the scene? is it an indoor or an outdoor environment? what was the temporal behavior in the scene? what elements of the system were tested in the environment? obviously the YOLO module cannot be evaluated in a simulated environment.
10. 4.2: To measure the accuracy of the SLAM module you need a reference. Please compare the SLAM results to a diagram of the surrounding environment and provide a metric for measuring accuracy.
11. p.13: What is "minicancanda 3"?
12. Please provide a link to a video the demonstrates the behavior of the algorithm in a real-world outdoor environment. That is, detection of obstacles, distances, SLAM results.
13. Please provide additional details regarding the camera. Field of view, resolution, typical angle relative to the ground.
14. Have the authors considered the detection of objects based on the LiDAR 3D data?
Comments on the Quality of English LanguageIt is suggested to make the manuscript shorter, specifically shortening the introductory part and presenting the results in a more concise manner.
Round 2
Reviewer 2 Report
Comments and Suggestions for Authors
The reviewer acknowledges the authors' efforts in providing the revised version of the manuscript.
Comments on the Quality of English LanguageThe manuscript can be made shorter and more concise. However no special comments on the quality of English.